# Increased Use of Porch or Backyard Nature during COVID-19 Associated with Lower Stress and Better Symptom Experience among Breast Cancer Patients

**DOI:** 10.3390/ijerph18179102

**Published:** 2021-08-28

**Authors:** Amber L. Pearson, Victoria Breeze, Aaron Reuben, Gwen Wyatt

**Affiliations:** 1Department of Geography, Environment and Spatial Sciences, Michigan State University, East Lansing, MI 48824, USA; breezevi@msu.edu; 2Department of Psychology, Duke University, Durham, NC 27708, USA; aaron.reuben@duke.edu; 3College of Nursing, Michigan State University, East Lansing, MI 48824, USA; gwyatt@msu.edu

**Keywords:** green space, parks, nature-watching, indoor nature, passive nature, stress, cancer symptoms

## Abstract

Contact with nature has been used to promote both physical and mental health, and is increasingly used among cancer patients. However, the COVID-19 pandemic created new challenges in both access to nature in public spaces and in cancer care. The purpose of our study was to evaluate the change in active and passive use of nature, places of engaging with nature and associations of nature contact with respect to improvements to perceived stress and symptom experience among breast cancer patients during the pandemic. We conducted a cross-sectional survey of people diagnosed with breast cancer using ResearchMatch (*n* = 56) in July 2020 (the first wave of COVID-19). In this US-based, predominantly white, affluent, highly educated, female sample, we found that, on average, participants were first diagnosed with breast cancer at 54 years old and at stage 2 or 3. Eighteen percent of participants experienced disruptions in their cancer care due to the pandemic. As expected, activities in public places significantly decreased as well, including use of parks/trails and botanical gardens. In contrast, spending time near home, on the porch or in the backyard significantly increased. Also observed were significant increases in indoor activities involving passive nature contact, such as watching birds through a window, listening to birdsong, and smelling rain or plants. Decreased usage of parks/trails was significantly associated with higher stress (Coef = −2.30, *p* = 0.030) and increased usage of the backyard/porch was significantly associated with lower stress (Coef = −2.69, *p* = 0.032), lower symptom distress (Coef = −0.80, *p* = 0.063) and lower symptom severity (Coef = −0.52, *p* = 0.009). The most commonly reported alternatives to outdoor engagement with nature were watching nature through a window (84%), followed by looking at images of nature (71%), and listening to nature through a window (66%). The least commonly enjoyed alternative was virtual reality of nature scenes (25%). While outdoor contact with nature away from home decreased, participants still found ways to experience the restorative benefits of nature in and around their home. Of special interest in planning interventions was the fact that actual or real nature was preferred over that experienced through technology. This could be an artifact of our sample, or could represent a desire to be in touch with the “real world” during a health crisis. Nature contact may represent a flexible strategy to decrease stress and improve symptom experience among patients with cancer, particularly during public health crises or disruptions to cancer care.

## 1. Introduction

Despite improvements to cancer diagnosis, treatment, and prognosis, cancer remains a highly prevalent health condition, both worldwide and within the United States (US). Globally, in 2020, there were an estimated 19+ million new cancer cases and nearly 10 million deaths, with female breast cancer as the most commonly diagnosed cancer [1]. In the US, cancer is predicted to surpass heart disease as the leading cause of death within the next ten years, with new cases expected to increase nearly 45% from 1.6 million to 2.3 million annually by 2030 [2]. Cancer also remains a health condition marked by high stress, painful symptoms, and, depending on the type and severity of disease, diminished quality of life.

While conventional medicine, particularly pharmacological intervention, addresses symptoms and quality of life, most cancer patients find additional benefit from non-pharmacological interventions. A survey of breast cancer patients found that more than 80% turn to complementary, behavioral, and integrative therapies for additional symptom management [3]. Among such therapies, “green exercise” or outdoor physical activity in nature is gaining attention as a low-cost, flexible activity with emerging evidence of effectiveness for symptom and stress management across a wide range of health conditions. Physical activity in particular has been linked to improved quality of life among cancer patients [4] as well as improved overall survival in the context of breast cancer. The proposed mechanisms through which physical activity affects breast cancer incidence and survival include changes in adiposity, sex hormones, insulin resistance [5,6], and weight management [7].

In order to increase physical activity and reduce chronic stress on a population level, researchers and city planners are exploring features of the built environment that may promote healthy lifestyles, including access to urban green spaces such as parks [8,9,10,11]. Previous research suggests that access to green space may reduce health disparities through numerous lifestyle and environmental pathways, some of them particularly important for those diagnosed with cancer, such as reduced inflammation and improved immune function [12,13,14,15,16,17,18,19,20,21,22,23,24,25,26]. Among other things, parks serve as places to engage in “green exercise”, which has been shown to lower perceived stress and risk of chronic disease more than exercise indoors [27,28]. Multiple studies underscore the twinned health benefits of access to green space, which encourages engagement in physical activity, contributing to weight management as well as stress reduction, both of which reduce inflammation and may play a role in improving health outcomes [29,30,31,32,33].

In addition to the benefits of physical activity in green spaces, research indicates that passive exposure to green space (e.g., visual, as in the sight of plants and trees, and auditory, as in birdsong) may lower stress with or without exercise [34,35]. Visual greenery is thought to serve as a calming backdrop to promote recovery from stress. Laboratory evidence has shown an increase in parasympathetic nervous activity and lowered heart rate when viewing images of trees and grass [36,37] and a preference for more “natural” images [38] and sounds [39]. Similarly, laboratory [40] and cross-sectional research has shown that natural sounds reduce fear, encourage walking [41,42], lower stress [34], and enrich sleep and concentration [43]. In fact, a meta-analysis of the evidence of natural sounds on health concluded that birdsong and water sounds, specifically, decrease stress, pain, and annoyance while increasing positive affect and wellbeing [44]. Therefore, physical access to green spaces and/or passive contact with nature through auditory, olfactory or visual exposure (e.g., through an open window) may prove to be important in promoting health and quality of life, particularly for vulnerable groups such as cancer patients.

However, many adults experienced barriers to getting outdoors, visiting parks, and engaging in physical activity during COVID-19 lockdowns, particularly adults with compromised immune systems such as patients with cancer. The COVID-19 pandemic, therefore, provides a unique opportunity to better understand the potential influence of forced changes in nature-engagement behaviors on health and quality of life among those diagnosed with cancer. The focus of this study was both how these behaviors may or may not have changed during the pandemic, but also whether there were perceived effects and alternatives taken up by this population to cope during lockdown. Such information can inform potential physical activity and nature-based interventions for those with cancer specifically and for patients with other health conditions more generally. Importantly, this study also identified potential alternatives to engagement in outdoor activities in public places, which may be vital options for immune-compromised populations to experience nature, reduce stress, and manage symptom distress during crises such as a pandemic.

Given this background, the aims of this research (Figure 1) were to understand among breast cancer patients:how and to what extent engagement in outdoor physical activity, usage of parks, and at-home contact with nature changed during COVID-19 lockdowns;the effects of these changes on perceived stress and symptom experience;how physical activity behaviors changed, in order to understand the breadth of impact on activities in this population; andalternative indoor or near-home contacts with nature that were substituted for therapeutic outdoor physical activity in public places.

While other emerging research has shown an increase in outdoor activities during the pandemic [45], it was anticipated that outdoor physical activity and usage of parks had declined among cancer patients due to high levels of fear about transmission of the virus in this population [46]. Moreover, it was hypothesized that, in contrast, at-home activities increased. In theory, at-home activities replaced those occurring in public spaces, again to prevent transmission. Building on evidence that engagement in outdoor physical activity is associated with lower stress compared to the effects of indoor physical activity [27,28] and that usage of parks is associated with better mental health outcomes [27,28], a further hypothesis was that decreases in outdoor physical activity and usage of parks were associated with poorer symptom experience and higher perceived stress. In contrast, increased at-home nature contact was hypothesized to be associated with better outcomes, as these activities are theorized to be alternatives to engagement with nature in public spaces.

## 2. Methods

This cross-sectional design study enrolled people with breast cancer, accessed via Research Match https://www.researchmatch.org/researchers/ (accessed on 17 June 2021). A RedCap survey was then sent to eligible participants based in the US, including a consent form (MSU IRB STUDY00004593), who were 18 years or older on 17 June 2020. From 17 June to 27 July a total of seven Research Match “searches” for possible participants were performed, followed by recruitment invitation e-mails. Follow-up reminder emails were sent weekly from 7 July to 30 July. In total, 1049 people from Research Match’s database were contacted, 76 (7.24%) of which indicated they were interested participating, and 59 (5.62%) of which partially completed the survey, with 56 having full data. This national sample was predominately well-educated, white and female, with a mean age of 63 years. The survey was registered with PHENX (https://www.phenxtoolkit.org/ (accessed on 27 August 2021)).

### 2.1. Survey Instrument

The survey was investigator-developed, covering: (1) to what degree (e.g., mildly, moderately, completely) and how the locations of active outdoor and passive contact with nature changed (e.g., walking in parks versus use of backyard); (2) why behaviors changed; (3) how physical activity behaviors changed, to understand the breadth of impacts on activities in this population; and (4) what alternative indoor or near-home contact with nature was substituted for therapeutic outdoor PA in public places. In addition, demographic details were obtained, including age, sex, race/ethnicity, age at diagnosis, stage at diagnosis, household composition, income, education, current treatment, and COVID-19 status if they had been tested.

Specifically, for the time period preceding COVID-19 lockdowns and the past month following lockdowns, the survey included questions on types, locations and frequencies of outdoor PA activities, and types, locations and frequencies of passive contact with nature. Questions about frequency of physical activity and nature engagement behaviors were asked on a Likert-type scale from 5 (very often or frequently) to 1 (never) for two time points (last month during the pandemic and in a typical month before the pandemic). To calculate changes in behaviors, the past frequency was subtracted from the current frequency, so that positive values indicated an increase in the behavior and negative values indicated a decrease in the behavior. Data on whether participants felt they had access to neighborhood public spaces and/or fitness facilities over the past week were also collected, as this may have necessitated changes in the behaviors of interest. These were phrased as statements (e.g., I can access/use parks) and were anchored on a 5-point Likert-type scale from strongly agree to strongly disagree. These two variables were re-coded to agree or disagree. Neutral values were coded as disagree.

Attitude toward nature was assessed using the Nature-relatedness 6 (NR-6) scale [47]. This scale has been shown to demonstrate good internal consistency, temporal stability, and to predict happiness, environmental concern, and nature contact [47]. It involves six questions about self-identification with nature, connectedness or spirituality, awareness or knowledge about nature or local wildlife, and the need for/comfort in nature. Each item was measured on a 5-point Likert scale (1 = strongly disagree, 5 = strongly agree). Average scores were then calculated where higher scores indicated stronger nature-relatedness.

Perceived stress was measured using Cohen’s 10-item Perceived Stress Scale [48], which rates items such as “In the last month, how often have you felt that you were unable to control the important things in your life” and “how often have you felt difficulties piling up so high that you could not overcome them”. Items were measured on a 5-point scale ranging from 0 (Never) to 4 (Very Often). Higher scores indicated greater stress. Reference values are 0–13 (low stress), 14–26 (moderate), and 27–40 (high).

Common cancer-related symptoms and symptom severity was measured using the MD Anderson Symptom Inventory (MDASI) [49], which was updated to include symptoms experienced by patients undergoing cancer treatments [50], including fatigue, pain, nausea, etc. This instrument has established reliability and validity in samples of patients with cancer [51]. Participants rated each symptom’s presence and greatest severity on an 11-point scale, from 0 (“not present”) to 10 (“as bad as you can imagine”) [51]. A mean core symptom score was calculated across the 13 core symptoms [50,51,52]. The MDASI also contains six items that describe to what degree the symptoms have interfered with different aspects of the patient’s life in the previous 24 h: general activity, mood, walking ability, normal work (including work outside the home and housework), relations with other people, and enjoyment of life. Each interference item is also rated on an 11-point scale from 0 (“does not interfere”) to 10 (“completely interferes”). A total symptom distress score was obtained by adding the score for all ten symptoms (range 0–60). The mean of all of these symptom interference items was then used as a measure of overall symptom distress (range 0–10).

Alternatives to outdoor engagement with nature were captured in two ways. First, a series of statements were queried for level of agreement. For example, “I enjoy watching nature through a window.” Responses were measured on a 5-point Likert-type scale from strongly agree to strongly disagree. These were re-coded into agree versus disagree. Neutral values were coded as disagree. Second, the survey included open-ended questions about alternatives and substitutions for nature experiences.

### 2.2. Statistical and Qualitative Data Analyses

First, to quantify the nature and extent of behavior change during the pandemic, change in frequency of outdoor physical activity and passive engagement with nature by location (home versus public spaces) were calculated and tested for significant differences in average frequencies before versus during the pandemic using paired *t*-tests (results shown graphically). Next, to estimate the effects of these changes on perceived stress and symptom experience, a series of linear regression models were fitted to test for associations between change in activities (for those where significant changes were observed) and each outcome (stress, symptoms). Models were adjusted for age, marital status, income, and stage at diagnosis. Due to the small sample size, bootstrap estimation of the standard errors were used in all regression modeling. Statistical analyses were completed in Stata v16 (Statacorp, College Station, TX).

Subsequently, the percentage of the sample which agreed that they enjoyed alternatives to outdoor engagement with nature, stratified by high (≥5) versus low symptom distress, was calculated. Next, open-ended questions were coded by similarity to form themes and report frequencies by demographic characteristics. Alternative nature-contact behaviors were noted that were used to substitute for therapeutic outdoor physical activity in public places, paying careful attention to previously identified human–nature interactions. Specifically, tallies of reports of indoor or near-home alternatives to previously identified common non-exclusive nature-interaction patterns [53] including encountering wildlife, walking to destination spots in nature, gazing out at landscapes, and walking with dogs were generated. These alternatives primarily involved interacting with digital nature (e.g., watching nature shows), utilizing private property to interact with nature more, going on longer or more frequent walks outside the home, and attempting to be “more observant” of nature around them.

## 3. Results

### 3.1. Demographics

This was a sample of predominantly female (98%), white (88%), high income (59% earned $60,000+/year), and educated (54% with a graduate/professional degree) breast cancer patients (Table 1). Most participants were married (55%) without children living at home. On average, participants were 63 years old and were first diagnosed with breast cancer at 54 years old. The majority of participants were stage 2 or 3. Most participants had a unilateral (82%) and/or hormone receptor positive (66%) diagnosis. For treatment, 89% underwent surgery and/or radiation (61%). Approximately 18% of participants experienced disruptions in their cancer care due to the pandemic. A total of 11% had received a positive COVID-19 test at the time of participation, and all these participants were in the high-income category. Average nature-relatedness score was 3.7 (sd = 0.7), which matches most older adults surveyed in other settings. For comparison, in seven surveys involving about 1000 people, average scores ranged from approximately 3.0 to 3.5 [47]. Average perceived stress score was 25.2 (sd = 7.5), typically considered to be the higher end of “moderate” stress. Average overall symptom distress was 3 (sd = 2.5) and symptom severity 2.2 (sd = 1.7) on scales where scores at or above 5 are considered moderate to severe [54]. A clear gradient in perceived stress, overall symptom distress, and symptom severity across income categories was observed, whereby higher-income participants tended to report better status in all outcomes. In contrast, nature-relatedness was highest in the low-income group, which was the smallest sample subgroup (9% of the sample).

### 3.2. Changes in Engagement in Outdoor Physical Activity, Usage of Parks, and at-Home Contact with Nature during COVID-19 Lockdowns among Breast Cancer Patients

As predicted, activities in public places significantly decreased during the pandemic (Figure 2). Frequency using parks/trails and botanical gardens both significantly decreased during the pandemic. In contrast, spending time near home, on the porch, or in the backyard significantly increased. Also observed were significant increases in indoor activities involving passive nature contact, such as watching birds through a window, listening to birdsong, and smelling rain or plants. Visiting indoor spaces such as gyms decreased. Only one indoor activity, looking at greenery, did not significantly change. Among activities with low frequency both before and during COVID-19 were riding a bicycle for leisure and visiting botanical gardens. When looking at perception of access that may have influenced changes in the behaviors, it was found that nearly 95% of participants felt they could access neighborhood public spaces (e.g., sidewalks) in the past week, whereas only 27% felt they could access fitness facilities (data not shown in tabular form).

### 3.3. Effects of These Changes on Perceived Stress Score (PSS) and Symptom Experience

Decreased usage of parks/trails was significantly associated with higher stress (Coef = −2.30, *p* = 0.030) and increased usage of the backyard/porch was significantly associated with lower stress (Coef = −2.69, *p* = 0.032) (Table 2). Across most models, age showed marginal significant associations, whereby younger participants had higher PSS scores. Being married/partnered was associated with higher stress. There was a clear income gradient across all models, whereby lower income was associated with much higher PSS scores than higher income.

For symptom experience, it was found that increased use of the backyard/porch was associated with lower distress (Coef = −0.80, *p* = 0.063) (Table 2). Cancer stage at diagnosis was significantly positively associated with distress across all but one model. Again, a clear income gradient was observed, whereby lower income was associated with higher distress. It was also found that increased usage of the backyard/porch was associated with lower symptom severity (Coef = −0.52, *p* = 0.009), decreased usage of botanical gardens was associated with higher symptom severity (Coef = −0.51, *p* = 0.049), and increased usage of gyms was associated with higher symptom severity (Coef = 0.36, *p* = 0.043). The same income gradient was observed. Married or partnered participants and those with higher stages of cancer at diagnosis had higher symptom severity, both of which were significant in most of the models.

### 3.4. Alternative Indoor or Near-Home Contact with Nature Substitutes for Therapeutic Outdoor Physical Activity in Public Places

It was found that across all alternatives, those with high symptom distress reported higher agreement with enjoyment of each alternative, compared to those with low symptom distress (Table 3). The most commonly enjoyed alternatives were watching nature through a window (84%), followed by looking at images of nature (71%), and listening to nature through a window (66%). The least commonly enjoyed alternative was virtual reality of nature scenes (25%).

When tallying open-ended responses involving alternatives, several themes emerged: (i) digital nature; (ii) use of private spaces; (iii) domestic pets; (iv) safe public spaces; (v) passive nature; and (vi) complementary therapies. Digital nature included the Internet and TV programs on nature, and was considered a poor substitute for nature, but was reported by four participants. One participant stated, “There is no substitutions for being outdoors. You can look at the window or at pictures in the internet; it just isn’t the same.” The use of private outdoor spaces included the home environment or near-home areas in the neighborhood. Ten participants stated that this was their alternative during the pandemic. Participants reported using their backyard, front porch, back deck, and neighborhood. Activities within these spaces included spending time with animals, doing yard work/gardening, reading, sitting/watching, and spending time with grandchildren. Domestic pets were considered an alternative by four participants, and all four cited active time spent with dogs, primarily outdoors. Three participants reported using outdoor public spaces they considered safe, including the neighborhood and places they felt they could remain socially distant. Participants reported both passive olfactory and visual enjoyment of nature from indoors, including opening windows for fresh air and watching bird feeder through the window. Complementary therapies featured for ten participants, and included yoga, stretching and indoor exercise, and meditation, handcrafts, and woodworking.

## 4. Discussion

This small study of changes in nature contact among mostly high-income breast cancer patients during the COVID-19 pandemic produced two primary findings. First, patients were found to have changed their outdoor physical activity and contact with nature in several ways, including by spending more time at home or near home (e.g., backyard or porch) and enjoying more passive contact with nature. This was not surprising, as immune-compromised individuals, such as breast cancer patients, were especially likely to adhere to recommendations from the Centers for Disease Control and Prevention for avoiding crowded locations. Indoor and passive nature-contact activities increased, including watching birds through a window, listening to birdsong, and smelling rain or plants. Interestingly, looking at greenery through the window did not significantly change. Given changes to the auditory environment (e.g., less traffic), perhaps birds became more noticeable for participants, whereas greenery was less changed by COVID-19 conditions (in the short term at least).

These findings were similar to those found in a majority white, high-income sample from Vermont during the pandemic, showing that participation in nature-related physical activity increased and was associated with income, sex, and employment [45]. Likewise, a survey among educated, older participants found that people visited nature more often during the pandemic [55]. In contrast, in a global study with respondents from 97 countries, restrictions related to COVID-19 resulted in reductions in nature-related leisure activities (e.g., birding) [56], and data from a low-income minority sample found decreased contact with nature and/or outdoor physical activity [57].

Second, patients who reduced their use of community outdoor spaces such as parks, trails, and botanical gardens experienced higher stress and cancer symptoms, while those who increased at-home nature contact experienced the opposite trend. These findings are supported by the growing literature indicating the therapeutic effects of active engagement with nature, particularly for those with cancer [58]. Nature may help cancer patients transcend their diagnosis for a period of time and experience respite from their stressful emotions and symptoms [59,60]. It was found that increased usage of the backyard/porch was significantly associated with lower stress, distress and symptom severity. Some of these activities could be more active (e.g., playing with the dog), but some participants reported more passive activities such as sitting, reading, and watching. Echoing the current study’s findings, a European study found that having green-blue nature views from the home was associated with fewer symptoms of depression and anxiety [61], and in Bulgaria, visible greenery from the home was associated with reduced depressive/anxiety symptoms [62]. Nature-based engagement in a private green space [63], near home [56], in local green spaces [64], or even within the home (e.g., houseplants) [62] have all been shown to support mental health in other global research during the pandemic. In the current study, some of the more passive forms of nature contact were the most commonly reported alternatives to outdoor nature experiences, and included indoor bird watching, listening to birdsong, and smelling rain or plants. Importantly, a clear income gradient was observed across all distress and stress models and some severity models. Income inequalities in perceived stress and symptom experience with cancer cannot be overlooked, particularly during crises such as a pandemic.

### Strengths and Limitations

This study has several limitations. First, the COVID-19 survey tool was developed rapidly without time for validation, due to the acute nature of the pandemic. Refining this tool for future use is important. The study was also small, although fairly large for a study of cancer patients conducted during a public health crisis with implications for those with compromised immune systems. All measures were self-reported and future work may be strengthened by consideration of biomarkers for objective findings as well, although perceived stress and cancer symptoms are the primary concern of complementary interventions in this domain. Nevertheless, hair or saliva cortisol would enhance the stress data, along with comparable markers for symptoms. This study’s sample was well-educated and reported high income overall, limiting the generalizability of our findings to all populations of cancer patients. This sample population may have had more/better access to nearby nature in their neighborhood or at/near home, accommodating local passive or active nature experiences not afforded in many low-income neighborhoods. Due to the small sample size, the homogeneity of the sample and the cross-sectional design of this study, relationships detected in this study require further research and must be interpreted with caution. Greater sample diversity is needed to generalize our findings; however, this work provides a window into the nature contact experience of breast cancer patients during a pandemic, and how these experiences may reduce stress or improve symptom experience.

## 5. Conclusions

Cancer patients’ changes in contact with nature and natural spaces were associated with differences in perceived stress and symptom severity. The importance of green exercise in public space such as botanical gardens, parks, and trails to promote health and well-being was reaffirmed. At- or near-home nature activities became even more important during the pandemic in this population and were consistently associated with lower stress and improved symptom experience. Alternatives to green exercise such as television or computer-based nature scenes were considered inadequate substitutes. For breast cancer patients, these findings provide direction for future work around complementary health interventions that utilize nature. Interventions that incorporate public nature experiences and indoor real nature experiences may reduce cancer symptoms and stress during health crises and in ways that are flexible to changes in nature access and patient needs.

## Figures and Tables

**Figure 1 ijerph-18-09102-f001:**
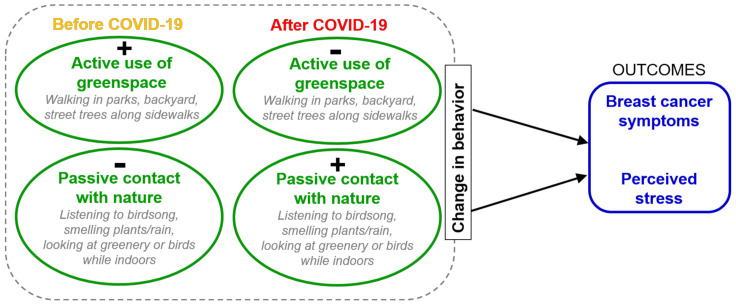
Conceptual diagram.

**Figure 2 ijerph-18-09102-f002:**
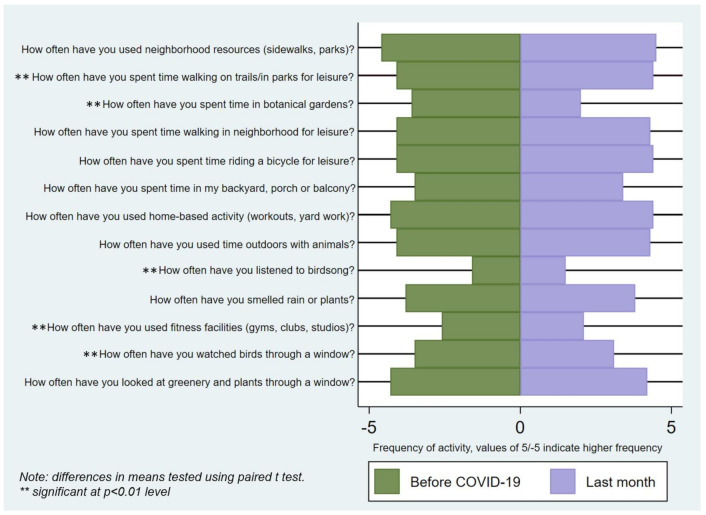
Changes in engagement in outdoor physical activity, usage of parks, and at-home contact with nature.

**Table 1 ijerph-18-09102-t001:** Demographic and health characteristics.

		Low Income	Middle Income	High Income	Total
		*n* = 5	*n* = 18	*n* = 33	*n* = 56
	Age (in years), mean (sd)	64.6 (11.6)	64.4 (12.3)	62.2 (9.9)	63.1 (10.7)
	Female, *n* (%)	5 (100)	18 (100)	32 (97.0)	55 (98.2)
	Married/partnered, *n* (%)	0 (0)	7 (38.9)	24 (72.7)	31 (55.4)
	Number of kids at home, mean (sd)	0 (0)	0 (0)	0.1 (0.3)	0.04 (0.3)
Ethnicity, *n* (%)	White	5 (100)	15 (83.3)	29 (87.9)	49 (87.5)
	Hispanic/Latino	0 (0)	1 (5.6)	0 (0)	1 (1.8)
	Black/African American	0 (0)	1 (5.6)	3 (10.0)	4 (7.1)
	Native American/American Indian	0 (0)	1 (5.6)	1 (3.0)	2 (3.6)
	Asian/Pacific Islander	0 (0)	0 (0)	0 (0)	0 (0)
	Other	0 (0)	0 (0)	0 (0)	0 (0)
Income, *n* (%)	Low: <$30,000/year				5 (8.9)
	Middle: $30,000–$59,999/year				18 (32.1)
	High: $60,000+/year				33 (58.9)
Education, *n* (%)	Less than college degree	3 (60.0)	6 (33.3)	2 (6.1)	11 (19.6)
	College degree	1 (20.0)	5 (27.8)	9 (27.2)	15 (26.8)
	Graduate/Professional degree	1 (20.0)	7 (38.9)	22 (66.7)	30 (53.6)
	Age at diagnosis, mean (sd)	55.0 (13.0)	55.2 (13.2)	52.9 (9.9)	53.8 (11.1)
	Stage at diagnosis, mean (sd)	2.8 (1.8)	2.6 (1.2)	2.6 (1.1)	2.6 (1.2)
	Experienced disruptions in care during COVID-19, *n* (%)	0 (0)	6 (33.3)	4 (12.1)	10 (17.9)
Diagnoses, *n* (%)	Bilateral	0 (0)	1 (5.6)	4 (12.1)	5 (8.9)
	Unilateral	3 (60.0)	14 (77.8)	29 (87.9)	46 (82.1)
	Recurrent	0 (0)	2 (11.1)	1 (3.0)	3 (5.4)
	Metastatic	1 (20.0)	3 (16.7)	6 (18.2)	10 (17.9)
	Inflammatory	0 (0)	1 (5.6)	1 (3.0)	2 (3.6)
	BRAC1 or BRAC2 genes	0 (0)	2 (11.1)	2 (6.1)	4 (7.1)
	Hormone receptor positive	2 (40.0)	11 (61.1)	24 (72.7)	37 (66.1)
	Hormone receptor negative	1 (20.0)	2 (11.1)	2 (6.1)	5 (8.9)
	Mother diagnosed with breast cancer	1 (20.0)	2 (11.1)	10 (33.3)	13 (23.2)
	Mother’s mother diagnosed with breast cancer	1 (20.0)	3 (16.7)	2 (6.1)	6 (10.7)
	Father’s mother diagnosed with breast cancer	0 (0)	0 (0)	1 (3.0)	1 (1.8)
Treatment, *n* (%)	Chemotherapy	3 (60.0)	7 (38.9)	14 (42.4)	24 (42.9)
	Radiation	3 (60.0)	15 (83.3)	16 (48.5)	34 (60.7)
	Surgery	4 (80.0)	16 (88.9)	30 (90.9)	50 (89.3)
	Hormonal therapy	3 (60.0)	12 (66.7)	17 (51.5)	32 (57.1)
	Targeted therapy	0 (0)	2 (11.1)	5 (15.2)	7 (12.5)
	Immunotherapy	1 (20.0)	1 (5.6)	1 (3.0)	3 (5.4)
	Complementary therapies (e.g., meditation)	1 (20.0)	2 (11.1)	5 (15.2)	8 (14.3)
	None	0 (0)	0 (0)	0 (0)	0 (0)
	Positive test for COVID-19, *n* (%)	0 (0)	0 (0)	6 (18.2)	6 (10.7)
	Nature-relatedness score, mean (sd)	4.0 (0.4)	3.7 (0.8)	3.7 (0.7)	3.7 (0.7)
	Perceived stress score, mean (sd)	31.6 (7.3)	26.6 (8.0)	23.5 (6.9)	25.2 (7.5)
	Overall symptom distress, mean (sd)	4.6 (3.5)	3.9 (2.6)	2.3 (2.0)	3.0 (2.5)
	Symptom severity, mean (sd)	3.8 (1.6)	2.5 (1.8)	1.9 (1.5)	2.2 (1.7)
	Quality of life-spiritual, mean (sd)	35.8 (4.9)	39.4 (9.9)	38.3 (10.4)	38.5 (9.8)
	Quality of life-fitness, mean (sd)	15.4 (3.2)	14.8 (6.7)	18.2 (5.0)	16.9 (5.6)

**Table 2 ijerph-18-09102-t002:** Regression modeling results: Predicting perceived stress score, overall symptom distress and symptom severity.

	Perceived Stress Score					Symptom Distress					Symptom Severity				
	Coef	se	95% CI		*p*-Value	Coef	se	95% CI		*p*-Value	Coef	se	95% CI		*p*-Value
Change in use of parks/trails	−2.30	0.86	−4.03	−0.57	0.010 **	−0.23	0.31	−0.85	0.39	0.454	−0.31	0.21	−0.73	0.10	0.136
Age	−0.15	0.08	−0.32	0.02	0.085 *	−0.01	0.03	−0.07	0.05	0.815	−0.02	0.02	−0.06	0.02	0.347
Married/partnered 2020	4.74	2.01	0.70	8.79	0.022 **	0.95	0.72	−0.50	2.40	0.195	0.70	0.48	−0.27	1.66	0.155
Low income	13.57	3.41	6.72	20.42	<0.001 **	3.05	1.22	0.59	5.51	0.016 **	2.72	0.82	1.08	4.36	0.002 **
Middle income	4.42	1.99	0.41	8.42	0.031 **	1.87	0.72	0.43	3.31	0.012 **	0.86	0.48	−0.10	1.82	0.078 *
High income	ref					ref					ref				
Stage breast cancer diagnosis	0.20	0.75	−1.31	1.71	0.793	0.48	0.27	−0.07	1.02	0.084 *	0.34	0.18	−0.02	0.71	0.062 *
Change in use of botanical gardens	−1.21	1.28	−3.77	1.35	0.346	−0.52	0.43	−1.38	0.34	0.232	−0.51	0.29	−1.08	0.06	0.081 *
Age	−0.13	0.09	−0.31	0.05	0.162	0.00	0.03	−0.06	0.06	0.984	−0.01	0.02	−0.05	0.03	0.576
Married/partnered 2020	5.09	2.13	0.81	9.37	0.021 **	0.97	0.71	−0.46	2.41	0.180	0.73	0.48	−0.23	1.69	0.131
Low income	12.51	3.59	5.29	19.72	0.001 **	3.05	1.20	0.63	5.47	0.015 **	2.67	0.80	1.05	4.28	0.002 **
Middle income	4.73	2.15	0.42	9.04	0.032 **	1.76	0.72	0.31	3.20	0.018 **	0.77	0.48	−0.19	1.74	0.115
High income	ref					ref					ref				
Stage breast cancer diagnosis	0.15	0.80	−1.46	1.75	0.856	0.48	0.27	−0.06	1.02	0.079 *	0.34	0.18	−0.01	0.70	0.059 *
Change in use of backyard or porch	−2.69	1.07	−4.85	−0.53	0.016 **	−0.80	0.37	−1.54	−0.06	0.034 **	−0.52	0.25	−1.03	−0.02	0.041 **
Age	−0.11	0.09	−0.29	0.06	0.191	0.00	0.03	−0.06	0.06	0.919	−0.01	0.02	−0.05	0.03	0.535
Married/partnered 2020	5.70	2.04	1.60	9.79	0.007 **	1.16	0.70	−0.24	2.56	0.103	0.86	0.47	−0.09	1.81	0.076 *
Low income	13.40	3.43	6.51	20.29	<0.001 **	3.27	1.17	0.92	5.63	0.008 **	2.77	0.80	1.16	4.37	0.001 **
Middle income	4.81	1.99	0.81	8.82	0.020 **	1.84	0.68	0.47	3.21	0.010 **	0.89	0.46	−0.04	1.82	0.061 *
High income	ref										ref				
Stage breast cancer diagnosis	0.54	0.78	−1.02	2.10	0.489	0.59	0.27	0.06	1.13	0.030 **	0.42	0.18	0.05	0.78	0.026 **
Change in bird watching	1.67	1.29	−0.92	4.26	0.200	0.26	0.44	−0.63	1.14	0.562	−0.25	0.30	−0.85	0.34	0.398
Age	−0.17	0.09	−0.35	0.01	0.07 *	−0.01	0.03	−0.07	0.05	0.745	−0.02	0.02	−0.06	0.03	0.438
Married/partnered 2020	4.84	2.12	0.57	9.11	0.027 **	0.94	0.73	−0.52	2.40	0.200	0.79	0.49	−0.19	1.78	0.113
Low income	11.20	3.62	3.92	18.48	0.003 **	2.76	1.24	0.27	5.25	0.031 **	2.68	0.84	0.99	4.36	0.002 **
Middle income	5.41	2.08	1.23	9.60	0.012 **	1.99	0.71	0.55	3.42	0.008 **	0.93	0.48	−0.04	1.90	0.060 *
High income	ref					ref					ref				
Stage breast cancer diagnosis	0.29	0.80	−1.32	1.90	0.719	0.49	0.27	−0.06	1.05	0.078 *	0.31	0.19	−0.07	0.68	0.104
Change in listening to birdsong	0.48	1.28	−2.08	3.04	0.709	−0.32	0.43	−1.18	0.54	0.462	−0.45	0.29	−1.02	0.12	0.121
Age	−0.15	0.09	−0.34	0.03	0.099 *	0.00	0.03	−0.06	0.06	0.941	−0.01	0.02	−0.05	0.03	0.554
Married/partnered 2020	5.04	2.16	0.70	9.37	0.024 **	1.05	0.73	−0.41	2.51	0.154	0.83	0.48	−0.14	1.81	0.090 *
Low income	12.03	3.62	4.75	19.31	0.002 **	3.01	1.22	0.56	5.45	0.017 **	2.67	0.81	1.04	4.29	0.002 **
Middle income	5.30	2.13	1.02	9.57	0.016 **	1.88	0.72	0.44	3.32	0.012 **	0.86	0.48	−0.10	1.82	0.077 *
High income	ref					ref					ref				
Stage breast cancer diagnosis	0.19	0.83	−1.47	1.86	0.817	0.42	0.28	−0.14	0.98	0.139	0.26	0.19	−0.11	0.63	0.162
Change in smelling rain or plants	−0.53	1.39	−3.33	2.27	0.704	−0.11	0.47	−1.06	0.84	0.818	−0.08	0.32	−0.72	0.56	0.804
Age	−0.14	0.09	−0.32	0.05	0.139	−0.01	0.03	−0.07	0.06	0.864	−0.02	0.02	−0.06	0.02	0.404
Married/partnered 2020	5.29	2.19	0.90	9.69	0.019 **	1.02	0.74	−0.46	2.51	0.173	0.77	0.50	−0.23	1.78	0.130
Low income	12.48	3.69	5.07	19.89	0.001 **	2.97	1.25	0.47	5.47	0.021 **	2.57	0.84	0.88	4.27	0.004 **
Middle income	5.08	2.13	0.81	9.35	0.021 **	1.93	0.72	0.49	3.37	0.010 **	0.95	0.49	−0.03	1.93	0.058 *
High income	ref					ref					ref				
Stage breast cancer diagnosis	0.14	0.80	−1.48	1.75	0.867	0.47	0.27	−0.07	1.02	0.089 *	0.34	0.18	−0.03	0.71	0.075 *
Change in use of gyms	1.17	0.74	−0.31	2.66	0.119	0.40	0.25	−0.11	0.90	0.119	0.36	0.17	0.03	0.69	0.034 **
Age	−0.13	0.09	−0.31	0.04	0.134	0.00	0.03	−0.06	0.06	0.929	−0.02	0.02	−0.05	0.02	0.451
Married/partnered 2020	5.97	2.16	1.62	10.31	0.008 **	1.27	0.73	−0.19	2.74	0.087 *	1.01	0.48	0.03	1.98	0.043 **
Low income	13.08	3.56	5.92	20.23	0.001 **	3.21	1.20	0.80	5.63	0.01 **	2.80	0.80	1.20	4.41	0.001 **
Middle income	4.85	2.07	0.69	9.01	0.023 **	1.84	0.70	0.43	3.24	0.011 **	0.86	0.46	−0.07	1.79	0.070 *
High income	ref					ref					ref				
Stage breast cancer diagnosis	0.13	0.78	−1.45	1.70	0.873	0.47	0.26	−0.06	1.00	0.082 *	0.34	0.18	−0.02	0.69	0.062 *

** Significant at *p* < 0.05 level; * significant at *p* < 0.10 level.

**Table 3 ijerph-18-09102-t003:** Alternatives to engagement with outdoor nature, by level of symptom distress.

	Low Symptom Distress (<5)	High Symptom Distress (≥5)	Total
	*n* = 44	*n* = 12	*n* = 56
I enjoy watching nature through a window.	36 (81.8)	11 (91.7)	46 (83.9)
I enjoy listening to nature through a window.	27 (61.4)	10 (83.3)	37 (66.1)
I enjoy looking at images of nature.	31 (70.5)	9 (75.0)	40 (71.4)
I enjoy listening to natural sounds through recordings (e.g., water, birds).	14 (31.8)	7 (58.3)	21 (37.5)
I enjoy growing indoor plants.	25 (56.8)	7 (58.3)	32 (57.1)
I enjoy virtual reality of nature scenes.	10 (22.7)	4 (33.3)	14 (25.0)

## Data Availability

Data available in Appendix A.

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
