# Peer review of "Increased Use of Porch or Backyard Nature during COVID-19 Associated with Lower Stress and Better Symptom Experience among Breast Cancer Patients"

_ijerph, 2021, doi:10.3390/ijerph18179102_

Round 1

Reviewer 1 Report

This paper aimed to examine possible changes in engagement with nature by breast cancer patients during Covid19 pandemic and the consequences of this to patients’ health. The main goal was to evaluate whether changes correlated to perceived stress and symptoms. Moreover, the study tried to identify what were alternative to outdoor engagement with nature, as a way of increasing knowledge of what may work as a buffer to stress and illness symptoms during health crisis. The authors concluded that indoor engagement with nature was, in some cases, correlated to stress reduction.

Overall, this study is a contribution to better understand the restorative contribution of nature engagement to health. Its relevance has mainly to do with addressing changes happening due to the pandemic situation and exploring alternatives that may work during such periods. The manuscript is well-written and presents clear arguments. Yet some aspects should be clarified by the authors, particularly regarding statistical analysis.

  • The keywords do not reflect the contributions of the study to health and stress management.
  • There is information missing about the participants - where are they from?
  • 144 and L.178 – why were these answers recoded to agree or disagree? What happened to middle-point scale answers?
  • The sample is very small, are regressions a good option for the analysis? Shouldn’t these analyses be supported by bootstrapping Confidence Intervals? These are available for regression analysis at SPSS software.

Author Response

We modified the words to better reflect the contributions of the study. We include the term parks because parks represent neighborhood amenities commonly used to engage in outdoor physical activity. We include green space as an alternative term for public spaces to engage in outdoor physical activity. We now also include: nature-watching (identified as an important alternative to outdoor activities), indoor nature (as an alternative to outdoor nature activities) and passive nature (as another term for indoor nature). We also include our outcomes of interest - stress and cancer symptoms.

Researchmatch is a database of USA-based participants. We now include this in the Abstract and in the Methods.

We now include this detail in those two locations – thanks for pointing out this omission. To be conservative, we coded neutral responses as disagree.

We too were concerned about the power to conduct certain statistical analyses. We used a few principles to guide this work. First, we included a limited number of independent variables in our regression models. Second, we observed a few clear trends across all models, indicating likely meaningful relationships worth reporting. We have taken your recommendation and now report bootstrapped standard errors, and related confidence intervals and p-values in the Results section and note this analytical change in the Methods. The overall conclusions for this study were not altered by these modified results.  

Reviewer 2 Report

The purpose of the research was to evaluate the change in active and passive use of nature, places of engaging with nature and associations of nature contact with improvements to perceived stress and symptom experience among breast cancer patients during the pandemic. It is relevant and interesting]

The topic is original because I have not read similar topics before. Moreover, I do not find similar research topics in Google Scholar. Three key elements are added to the subject area (environmental research and public health): use of porch or backyard nature, during COVID-19, breast cancer patients

It is well-written because few grammar mistakes are found. The text clear and easy to read

The conclusions are consistent with the evidence and arguments presented? They address the main question posed

We seldom use "Subject pronouns", e.g. "we" in reporting research results. The following sentence structure may be changed. The same sentence structure may be applied in reporting the research results. 

Original 1: "We, again, observed the clear income gradient, whereby lower income was associated with higher distress." (Lines 253-255) 

New 1: "It is observed the clear income gradient, whereby lower income was associated with higher distress." (Lines 253-255) 

Original 2: "We also found that increased usage of the backyard/porch and was significantly 255 associated with lower symptom severity (Coef=-0.52, p=0.041), decreased usage of botan-ical gardens was associated with higher symptom severity (Coef=-0.51, p=0.081), and in-257 creased usage of gyms was associated with higher symptom severity (Coef=0.36, p=0.034)." (Lines 255-258)

New 2: "It is also found that increased usage of the backyard/porch and was significantly 255 associated with lower symptom severity (Coef=-0.52, p=0.041), decreased usage of botan-ical gardens was associated with higher symptom severity (Coef=-0.51, p=0.081), and in-257 creased usage of gyms was associated with higher symptom severity (Coef=0.36, p=0.034)." (Lines 255-258)

Author Response

The text was altered to remove personal pronouns throughout.  

Reviewer 3 Report

This study explored that the increased use of porch or backyard nature during COVID-19 associated with lower stress and better symptom experience among breast cancer patients. The topic is interesting, results are valuable from both theoretical and practical perspectives. However, there are some questions needing to be addressed. My detailed suggestions are as follows:

  • For the introduction: at the beginning, authors presented the cancer condition in US, however, International Journal of Environmental Research and Public Health is an international journal, it’s for global readers, thus, I suggested that authors should present the condition of cancer from global perspective. In the hypothesis introduction part, the author directly leads to the specific hypothesis, and the logic is poor. Thus, authors need spend more words to demonstrate the hypothesis.
  • For the method, it is better for authors to move the demographic information (such as sex, age, race, and socioeconomic status) from the result section to the participants’ section so that the reader can better understand the subject information.
  • For the results section: authors describe changes in the exposure of patients to nature during COVID-19, however they did not test changes significant or not, thus, a one sample t test is recommendation. The effects of various natural contacts on physical and mental health were analyzed respectively in Section 3.2. However, there may be some collinearity among various natural contacts. Therefore, it is suggested to analyze the effects of natural contacts on patients' physical and mental health from an integrated perspective.
  • For the discussion section, the interpretation of the relationship between natural contact and physical and mental health need more cautious, after all, it is a cross-sectional study. The patients who were willing to participate this study, are the white and high social class group, and the reasons need further explanation.

Author Response

The Introduction was edited to provide global statistics. We kept the US-specific statistics as well, as this is the population from which our sample was drawn. In the Introduction section, after the research aims, we now include text to describe the rationale for the hypotheses, including citations of complementary research.

We have added brief demographic information about the sample in the Methods section. We retained the in-depth descriptive statistics in the Results section as well, as many readers from public health may expect details in this section.  

We added the significant results of the paired t tests to Figure 1 – thank you. Among the behavior change variables, we observed three moderate/strong correlations (shown below). 1) Birdwatching and listening to birdsong; 2) Birdwatching and walking in the neighborhood for leisure; and 3) Walking in the neighborhood and using neighborhood resources. With modest correlations between most variables, we feel that that our regression analysis approach is reasonable. [see attachment]

We agree that this study’s findings must be interpreted with caution. We reviewed all text in the Discussion to ensure causality was not implied. In addition, we added text to the Limitations section related to the cross-sectional design.  

Round 2

Reviewer 3 Report

The manuscript had been improved, however, there were two issues which were not responsed.

  • authors describe changes in the exposure of patients to nature during COVID-19(line 244-line255), however they did not test changes significant or not, thus, a one sample t test is recommendation.
  • The effects of various natural contacts on physical and mental health were analyzed respectively in Section 3.2. However, there may be some collinearity among various natural contacts. Therefore, it is necessary to analyze from an integrated perspective. As a reader or practitioner, I want to a panorama.

Author Response

Below I respond to each item.

  • authors describe changes in the exposure of patients to nature during COVID-19(line 244-line255), however they did not test changes significant or not, thus, a one sample t test is recommendation.
  • RESPONSE:  For the analyses in Fig 2, we used a paired t test to evaluate significant differences within the same individuals before and during COVID-19. Because we were testing for differences in the same population at two time periods, we used a paired t test.
  • The effects of various natural contacts on physical and mental health were analyzed respectively in Section 3.2. However, there may be some collinearity among various natural contacts. Therefore, it is necessary to analyze from an integrated perspective. As a reader or practitioner, I want to a panorama.
  • RESPONSE: We provided a correlation matrix for all of the nature contact/physical activity variables in our last response and noted that most were modest, thus, minimizing the concern over collinearity. We opted not to fit a regression model with several of the nature contact/physical activity variables at one time (in addition to the demographic variables) because of the small sample size. If the reviewer is requesting another type of analysis, please let us know specifically what is recommended.